# Glioblastoma Microenvironment and Invasiveness: New Insights and Therapeutic Targets

**DOI:** 10.3390/ijms24087047

**Published:** 2023-04-11

**Authors:** José Ignacio Erices, Carolina Bizama, Ignacio Niechi, Daniel Uribe, Arnaldo Rosales, Karen Fabres, Giovanna Navarro-Martínez, Ángelo Torres, Rody San Martín, Juan Carlos Roa, Claudia Quezada-Monrás

**Affiliations:** 1Laboratorio de Biología Tumoral, Instituto de Bioquímica y Microbiología, Universidad Austral de Chile, Valdivia 5090000, Chile; 2Millennium Institute on Immunology and Immunotherapy, Universidad Austral de Chile, Valdivia 5090000, Chile; 3Department of Pathology, School of Medicine, Pontificia Universidad Católica de Chile, Santiago 8330024, Chile; 4Millennium Institute on Immunology and Immunotherapy, Pontificia Universidad Católica de Chile, Santiago 8331150, Chile; 5Escuela de Medicina Veterinaria, Facultad de Recursos Naturales y Medicina Veterinaria, Universidad Santo Tomás, Talca 8370003, Chile; 6Laboratorio de Patología Molecular, Instituto de Bioquímica y Microbiología, Universidad Austral de Chile, Valdivia 5090000, Chile

**Keywords:** glioblastoma, glioblastoma stem-like cells, tumor microenvironment, tumor infiltration, invasiveness, brain tumors

## Abstract

Glioblastoma (GBM) is the most common and malignant primary brain cancer in adults. Without treatment the mean patient survival is approximately 6 months, which can be extended to 15 months with the use of multimodal therapies. The low effectiveness of GBM therapies is mainly due to the tumor infiltration into the healthy brain tissue, which depends on GBM cells’ interaction with the tumor microenvironment (TME). The interaction of GBM cells with the TME involves cellular components such as stem-like cells, glia, endothelial cells, and non-cellular components such as the extracellular matrix, enhanced hypoxia, and soluble factors such as adenosine, which promote GBM’s invasiveness. However, here we highlight the role of 3D patient-derived glioblastoma organoids cultures as a new platform for study of the modeling of TME and invasiveness. In this review, the mechanisms involved in GBM-microenvironment interaction are described and discussed, proposing potential prognosis biomarkers and new therapeutic targets.

## 1. Introduction

Glioblastoma (GBM) is the most frequent and deadly malignant brain tumor in adults, with an estimated incidence in the United States of 3–5 cases per 100,000 person-years [1]. Currently, GBM treatment consists of multimodal therapy, which includes surgery, radio- and chemotherapy, temozolomide (TMZ) being the gold standard drug [2]. Despite multimodal therapy, the average patient survival does not exceed 15 months and only 26% of patients live up to 2 years post-treatment [2]. The low efficacy of GBM therapies is mainly caused by tumor cell infiltration into the healthy brain tissue, which makes the total surgical tumor resection challenging, impairs the localized effect of radiotherapy and leads inevitably to GBM’s recurrence with fatal outcomes [3]. The GBM infiltration phenotype is considered one of the main characteristics associated with the therapy failure, so understanding its cellular and molecular mechanisms will help to develop new therapeutic strategies to decrease GBM’s aggressiveness. GBM cells are found in particular niches, characterized by extracellular matrix (ECM) interaction, low oxygen pressure, tumor-associated soluble factors and different cell types composing the tumor microenvironment (TME), which is strongly linked to the modulation of the GBM invasive phenotype [4,5]. This review compiles more than 100 studies on the role of several cellular and non-cellular components of the TME that regulate the invasive phenotype and aggressiveness of GBM. Furthermore, we review different protocols for patient-derived glioblastoma establishment, as well as a wide range of assays for modeling of the tumor microenvironment and invasiveness.

## 2. Cellular Invasion Mechanisms

Despite non-brain tumor cells invading organs through the circulatory system or lymph, GBM cells are considered non-metastatic to extra-cranial organs; however, they perform local invasion into the healthy brain tissues [6]. GBM cells are able to infiltrate through perivascular space around blood vessels or between neurons and glia [7,8]. The cell infiltration into the brain parenchyma requires the activation of cellular processes such as interaction and degradation of components of the ECM, remodeling of the cytoskeleton, and changes in cell volume [7]. Various tumor cell movement patterns have been described, all of which allow the cells to invade the surrounding tissue in a single-cell movement manner, which comprises mesenchymal or amoeboid movement, and collective movement by the so-called “cluster or strand” [9]. Single-cell movement is the main cellular infiltration mechanism responsible for tumor recurrence and, surprisingly, the same movement pattern is the one used by neural stem cells (NSCs) during the embryonic nervous system development and the response to brain tissue damage [10]. In recent years, emerging evidence has suggested that GBM cell invasion mechanisms are not only dependent on inherent cell characteristics but are also strongly regulated by dynamic communications and interactions between tumor cells and their TME [11].

## 3. Tumor Microenvironment

Most aggressive tumors, such as GBM, can modify their microenvironment, favoring the development of tumorigenic properties, such as chemoresistance, cell proliferation, migration, and invasion [11]. GBM TME components are both cellular (e.g., glioblastoma stem-like cells, endothelial cells, microglia, astrocytes, and neurons) and non-cellular (e.g., ECM, variations in hypoxia levels and soluble molecules) [12]. Here, we discuss how these different TME components participate in promoting the GBM invasive phenotype.

### 3.1. Cellular Components Involved in GBM Cell Invasion

#### 3.1.1. Glioblastoma Stem-Like Cells

One feature of GBM is its high cellular heterogeneity, which has been used as a prognostic indicator [13]. The presence of cells with self-renewing and multi-lineage differentiation properties called glioblastoma stem-like cells (GSCs), has been proposed as the main cause of tumor initiation, growth, and recurrence during the progression of GBM [14]. Nowadays, GSCs are identified with the use of biomarkers associated with the NSC membrane protein, such as CD133 and CD44 [14]. GBM cell subtypes enriched with CD133 display an increased migratory and invasive capacity, resembling molecular profiles described for angiogenesis and cell invasion [15]. Yu et al. showed that GSCs CD133(+) have a more migratory and invasive phenotype than GSCs CD133(−) both in vitro and in vivo [16]. Organotypic rat brain slices inoculated with GBM CD133(+) cells showed a greater infiltration of healthy brain tissue, mainly through perivascular niches and white matter tracts, than GBM CD133(−) cells [16]. Nishikawa et al. reported CD44 as another GBM biomarker in tumors with exacerbated migration and invasion pathways, preferentially at the tumor periphery [17]. CD44 knockdown inhibited GSCs’ migration and invasion both in vitro and in vivo [17]; mouse brain tumors generated from CD44-knockdown GSCs were less invasive and mice survived significantly longer than control mice [18]. Furthermore, Cheray et al. identified that *KLRC3*, the gene coding for NK cell group 2 isoform E (NKG2E) protein, is overexpressed in glioblastoma undifferentiated cells compared to the differentiated ones and that its silencing decreases the GSCs’ invasion [19]. Other groups have reported an upregulation of proteins involved in the migration and invasion of GSCs, such as matrix metalloproteinases (MMPs), different members of adamalysins including ADAMs (A disintegrin and metalloproteases) and ADAMTS (A disintegrin-like and metalloprotease domain (reprolysin type) with thrombospondin type 1 motifs), and adhesion receptor proteins (integrins) [20]. Alonso et al. observed that the ectopic expression of the transcription factor SOX2 was essential to induce and maintain GSCs’ migration and invasion [21]. In addition, researchers have measured high expression levels of stem cell markers in the tumor frontal invasion border, supporting the idea of GSCs as being responsible for the highly invasive phenotype of GBM [22]. Several signaling pathways, such as Wnt (wingless-INT), TGF-β (transforming growth factor-beta) and hedgehog (Hh)-GLI1 (glioma-associated oncogene homolog 1) pathways have been reported as determinant modulators of the GBM invasive phenotype [23]. Impaired Wnt signaling is related to GBM’s poor prognosis and its aberrant activation is implicated in the maintenance of a highly invasive phenotype [24]. The intranuclear localization of β-catenin has been reported at the infiltrating edge of tumors while the activation of the canonical Wnt/β-catenin pathway maintains the stemness and enhances GSCs’ mobility through ZEB1 (zinc finger E-box-binding homeobox 1) activation [25]. Likewise, Wnt5a, a non-canonical Wnt ligand, is overexpressed in high-grade GBM and GSCs being involved in GBM infiltration by regulating MMP-2 expression [26]. In addition, Binda et al. demonstrated that highly infiltrating mesenchymal glioblastoma cells were associated with the expression of Wnt5a and that its overexpression induces an invasive phenotype and activates the typical cell invasion genes in low-invading GSCs [27]. Furthermore, TGF-β signaling promotes GBM cells’ stemness through SOX4-SOX2 pathway activation, and Smad-dependent induction of LIF (leukemia inhibitory factor), which subsequently activates the Janus kinase (JAK)-signal transducer and activator of the transcription (STAT) pathway promoting the migration and invasion of GSCs [28,29]. TGF-β signaling regulates alpha V beta 3 integrin (vitronectin receptor), MMPs, and the tissue inhibitor of metalloproteinases (TIMP)-2 and cathepsin expression, which are relevant for ECM interaction and degradation during GSCs’ invasion [30]. The Hh signaling pathway is critical in GBM tumorigenesis, as well as in expressing transcription factor GLI1 [31]. Hh pathway activation promotes GSCs’ invasion and angiogenesis, thereby enhancing snail, slug, and vascular endothelial growth factor (VEGF) expression [31,32]. Strong evidence suggests the importance of GSCs in the infiltrative nature of GBM (Table 1). New therapeutic proposals can be designed to reduce the recurrence of patients diagnosed with brain tumors. Interestingly, it is not only the undifferentiated cells that affect the invasiveness of GBM, as cells that do not present malignant characteristics have a positive impact on the processes of migration and invasion of tumor cells (Figure 1).

#### 3.1.2. Endothelial Cells

Inside the tumor tissue, GSCs are in contact with endothelial cells (ECs), creating a perivascular niche, which impacts the GBM’s progression. The perivascular niche is identified with the presence of the molecular markers such as CD31, CD34, CD133, nestin, α-SMA (alpha smooth muscle actin), GFAP (glial fibrillary acidic protein), CD14, and CD44 [33]. GSCs’ infiltration can be promoted via VEGF secreted by ECs, which may induce the transdifferentiation of GSCs into ECs, promoting angiogenesis and invasiveness [34]. The recently developed on-chip platform has further permitted live-cell imaging of GSC–microvessel interaction, enabling quantitative analysis of GSC polarization and migration, showing the importance of VEGF released from ECs for the invasiveness of GSCs [35]. Other soluble factors, such as angiopoietin (Ang)-1, regulate the crosstalk between glioma cells and ECs. The signaling axis between tyrosine kinase receptor Tie2/TEK and Ang-1 promotes cell invasion by regulating the expression of adhesion proteins, such as N-cadherin and integrin β1 [36]. ECs have been reported to attract GBM cells to blood vessels through the release of bradykinin (BK), which can activate the signal transduction pathways for B1 and B2 bradykinin receptors present in glioma cells [37].

Montana et al. have demonstrated that BK contributes to cell migration by activating Ca^2+^ intracellular mobilization. B2 receptor knockdown has shown that the association between blood vessels and GBM cells is decreased in rat brain slices [37]. Yadav et al. indicate that the overexpression of C-X-C chemokine receptor type 4 (CXCR-4), also known as fusin or CD184, in primary cultures of human GSCs and mouse glioma cells exhibits significant migration towards human (HBMVE) and mouse (MBMVE) brain microvascular ECs [38]. CXCL12 appears to be the main factor regulating cell migration when cells are attracted by MBVE and HBMVE. Additionally, the knockdown of CXCR-4 in mouse glioma cells inhibits the in vitro migration towards MBVE cells, decreasing perivascular invasion [38]. Additionally, the intracranial inoculation of CXCR-4 knockdown glioma cells reduced tumor growth and perivascular invasion, resulting in more sensitivity to radiotherapy, leading to increased survival [38]. McCoy et al. reported in a co-culture of patient-derived GBM cells and ECs an increased GBM invasiveness due to ECs via interleukin (IL)-8. In addition, the intracranial co-injection of GBM with ECs in an orthotopic mice model increased the tumor volume and led to less localized and more widely spread tumor formation. Both the in vitro and in vivo blockage of the IL-8 signaling reversed the GBM growth and cell invasion induced by ECs [39]. In a different study, ECs (HMEC-1) secreted stromal cell-derived factor-1 (SDF-1, also known as CXCL12), increasing the expression of cysteine cathepsins (B and S), MMP-9, and downregulating the endogenous cell adhesion molecule NCAM (neural cell adhesion molecule, also called CD56), in this way enhancing the invasiveness in U87MG cells [40]. The SDF-1 neutralizing antibody blocked endothelial cell-enhanced invasion of U87 cells through the downregulation of MMP-9, which suggests that under normoxic conditions GBM cells may be attracted by ECs [40]. All together, these data indicate that ECs-related molecules are potential therapeutic targets to impair perivascular GBM invasion (Table 1 and Figure 1).

#### 3.1.3. Glial Cells

The GBM TME includes different non-cancer cells, such as astrocytes, oligodendrocytes, and microglia, that are able to support tumor growth [41] (Table 1). Astrocytes constitute about 50% of the human brain volume, having an important role in brain physiology and diseases [42]. The GBM cells can activate surrounding astrocytes (astrogliosis), which secrete high amounts of chemokines, such as IL-6, thereby enhancing GBM cell invasion and tissue infiltration by increasing the expression of MMPs [43]. In addition, GNDF (glial cell line-derived neurotrophic factor), secreted by astrocytes, induces the invasion of GBM cells by activating RET (rearranged during transfection)/GFRα1 (GDNF family receptor alpha-1) receptors and pro-tumoral signaling pathways, such as MAPK (mitogen-activated protein kinases) and PI3K (phosphatidylinositol 3-kinase)/Akt [44]. Edwards et al. reported that reactive astrocytes secrete CTGF (connective tissue growth factor), triggering NFkB (nuclear factor kappa B) signaling activation and the subsequent expression of ZEB1 in GBM cells, stimulating the epithelial–mesenchymal transition (EMT) and tumor cell infiltration [45]. This suggests that astrocytes are attracted to the GBM cells and facilitate their infiltration into the healthy brain tissue by the paracrine secretion of diverse molecules. Different studies describe that GBM cells are capable to release extracellular vesicles (EVs) into the tumor microenvironment. These tumor derived EVs are internalized by surrounding astrocytes, thus altering the expression and release of cytokines, which will promote tumor proliferation and invasion [46]. Furthermore, connexin 43 (Cx43), whose expression is elevated in glioma-associated astrocytes, is a major gap junction protein allowing direct communication between astrocytes and GBM cells [47]. Ectopic expression of a truncated Cx43 form altered the channel formation between GBM cells and astrocytes, decreasing GBM spreading into the brain parenchyma [48] (Figure 1).

Oligodendrocytes also play important roles in brain physiology, by regulating neuronal activities, neural plasticity, and metabolic support [49]. Oligodendrocytes are localized in the border of the tumoral microenvironment and are able to stimulate GBM cell invasion [50]. Kawashima et al. reported that oligodendrocyte cells overexpress Ang-2, upregulating U251 and T98G cell lines invasiveness [51]. In addition, it has been reported that dual blockade of Ang-2/VEGF may prolong survival of GBM patients by reprogramming the tumor immune microenvironment and delaying tumor growth [52].

Tumor infiltrating immune cells account for nearly 30% of all tumor cells, including macrophages (T cells, NK cells, and macrophages derived from bone marrow) [53]. Different types of lymphocytes are capable of infiltrating into the tumor tissue, mainly CD4+ T helper, CD8+ T cytotoxic, and Tregs [54]; where CD4+ presents a higher number than CD8+, being associated with a higher degree of aggressiveness, these cellular types are called tumor-infiltrating lymphocytes (TILs) [55]. TILs have been associated with specific alterations to the transcript profile of GBM. Based on the analysis of 171 histopathological images obtained from the TCGA database, the authors find a relationship between lymphocyte infiltration and the histopathological and mutation characteristics of GBM. It was observed that mesenchymal subtype GBMs have a higher number of TILs, unlike proneural and classic subtypes. Additionally, these TILs were mainly associated with tumors with mutations in the NF1 and RB1 genes [56]. In addition, the release of pro-inflammatory cytokines released by T cells has been shown to contribute significantly to the invasiveness of GBM cells, suggesting that TILs play an important role in maintaining the characteristics of mesenchymal GBM [57]. Revi and collaborators, from a multidimensional study that integrates the gene expression of immune cells from 12 GBM samples, showed that there is a subset of TILs that secrete IL-10, specifically in regions of the GBM with mesenchymal characteristics, contributing to the maintenance of the invasive capacity as well as the immunosuppressive microenvironment [58].

On the other hand, NK cells have been shown to play an important role in the negative regulation of GBM cell invasiveness. Studies in BALB/c-nude mice showed that inactivating NK cells led to an increase in widespread GBM metastasis [59]. GBM cells can inhibit the cytotoxic activity of NK cells by expressing MHC-I and/or PD-1, suggesting that these proteins may play an important role in promoting the invasion of GBM cells into healthy tissue [60,61]. The authors suggest NK cells play important inhibitory role over extracranial metastasis, this is possibly through direct interaction between GBM cells and NK cells, in areas where the immune cells monitor the abnormal cells [59].

Microglia are resident myeloid cells in the central nervous system (CNS) that control homeostasis and protect the CNS from damage and infections. Microglia and peripheral myeloid cells accumulate and adapt tumor-supporting invasiveness of GBM cells, through the release of several chemoattractants [62]. The activation of CX3CL1 (chemokine C-X3-C motif ligand 1, also known as fractalkine)/CX3CR1 (CX3C chemokine receptor 1) in tumor-associated microglia/macrophages (TAMs) increases the adhesion/migration capacity of GBM cells through the expression of MMP-2, -9, and -14 [63]. Markovic et al. studied the role of microglia in GBM invasion using brain slices ex vivo, and after removing the microglia with clodronate-filled liposomes, GBM cells were unable to infiltrate into healthy brain tissue, demonstrating the importance of this cell type in GBM invasion [64]. Furthermore, it has been reported that the microglia-GBM cells’ crosstalk modulates tumor infiltration by the activation of epidermal growth factor receptor (EGFR) and colony stimulating factor 1 receptor (CSF-1R) signaling [65]. Ye et al. reported GSCs CD133(+) became more aggressive after being co-cultured with TAMs; however, the GSCs’ invasion was inhibited by the neutralization of TGF-β1, decreasing the expression of MMP-9 in GBM cells [66]. Moreover, Wnt signaling is one of the main pathways related to tumor progression and cell invasion [67]. Wnt3a released by GBM cells can stimulate the activation of Wnt/β-catenin pathways in microglia cells, thereby inducing a protumor M2-like profile in these cells; this activation increases the invasion of the GBM cells [68]. Kloepper et al. reported using mice bearing orthotopic syngeneic (Gl261) GBM and human (MGG8) GBM xenografts and found that the dual blockade of Ang-2/VEGF reprogrammed the protumor M2 macrophages toward the antitumor M1 phenotype, improving the survival of GBM mice [69].

Historically, the activation of TAMs has been classified as the proinflammatory M1 and immunosuppressive M2 state; however, the use of single cell RNA sequencing (scRNA-seq) has allowed the identification of new molecular patterns in this cell type. One scRNA-seq study from human biopsies identified that CD11b+ TAMs were capable of simultaneously expressing M1 and M2 activation markers [70]. On the other hand, a study using flow cytometry and scRNAseq revealed that CD11b+ cells derived from xenografted tumors present a remarkable functional heterogeneity, in which the subpopulation of TAMs expressing Ccl22, Cd274 (encoding PD-L1), and Ccl5 supports a functional immunosuppressive state [71]. Another study based on scRNA-seq analysis of 20.1986 cells, in which are included glioma, immune, and stromal cells reported that immune cells infiltrating tumor tissue show extensive molecular heterogeneity, identifying nine myeloid subtypes, indicating that S100A4 expression is a regulator of immunosuppressive T and myeloid cells in GBM [72] Cui et al. using scRNAseq data analysis covering a total of 16.201 GBM-infiltrating immune cells, demonstrated the existence of new microglia subgroups (primed and repressed), which this subgroup varied according to GBM subtype [73], where the proneural subtype was characterized by primed microglia, and the classical repressed microglia [73]. Unfortunately, of the nine GBM tissues analyzed from which data were obtained, GBMs of the mesenchymal subtype were not included; however, due to the high intracellular heterogeneity of this malignant tumor, it has been shown that regions of GBM exhibit characteristics of the different subtypes, suggesting that there is a distinct immune cell infiltration [74]. Experimental evidence published by different research groups shows that the analysis of single cells in GBM is opening the possibility of identifying and understanding the plasticity of TAMs, clearly indicating the existence of new states in addition to M1 and M2, as initially understood.

**Table 1 ijms-24-07047-t001:** Cellular components of tumoral microenvironment involved in GBM invasiveness.

Source	Molecule	Mechanism	Reference
**GSCs**	CD133	Increased migratory and invasive capacity in GBM CD133(+) cell subtypes. More migratory and invasive phenotype in GBM CD133(+) cells inoculated in rat brain.	[15,16]
	CD44	Tumor cells CD44(+) have exacerbated migration and invasion pathways. CD44 knockdown inhibited GSCs’ migration and invasion both in vitro and in vivo.	[17,18]
	*KLRC3*	Silencing exhibited decrease in cell invasion.	[19]
	SOX2	Induce and maintain cell invasion and migration.	[21]
	Wnt5a	Involved in GBM infiltration by regulating MMPs’ expression. Overexpression is responsible for inducing the invasive phenotype and activation of cell invasion genes.	[25,26,27]
	LIF	Activation promotes cell migration and invasion.	[28,29]
	TGF-β	Signaling regulates MMPs and cathepsin expression, relevant in ECM degradation during cell invasion.	[30]
	Hh	Signaling promotes cell invasion, enhancing snail, slug, and VEGF expression.	[31]
**Endothelial cells**	VEGF	Promotes transdifferentiation, angiogenesis, and invasiveness of GSCs.	[34,35]
	Ang-1	Through Tie2 receptor induces GBM cell invasion by triggering expression of adhesion proteins.	[36]
	BK	Contributes to GBM cell migration by binding to B2 receptor and activating Ca^2+^ intracellular mobilization.	[37]
	CXCR4	Overexpression causes cell migration in GSCs and GBM cells, towards HBMVE and MBMVE.	[38]
	IL-8	Signaling increases GBM invasiveness to ECs.	[39]
	SDF-1	Increases invasiveness in GBM cells, expression of Cys cathepsins, and downregulates NCAM.	[40]
**Glial cells**	IL-6	Secreted by astrocytes; enhances GBM cell invasion and infiltration.	[43]
	GNDF	Secreted by astrocytes; induces GBM cell invasion by activating RET tyrosine kinase receptor, MAPK, and PI3K/Akt pathways.	[44]
	CTGF	Secreted by astrocytes; triggers NFKB signaling and subsequent expression of ZEB1, stimulating GBM cell infiltration.	[45]
	Cx43	Gap junction protein allowing direct communication between astrocytes and GBM cells.	[48,49]
	Ang-2	Overexpressed in oligodendrocytes; upregulates U251 and T98G cell lines’ invasiveness.	[51]
	CX3CL1/CX3CR1	Its activation in tumor-associated microglia/macrophages increases the adhesion/migration capacity of GBM cells through the expression of MMP-2, -9, and -14.	[63,64]
	CSF-1R	Active form released by microglia; enhances GBM cell invasion.	[65]
	Wnt3a	Released by GBM cells; activates Wnt/β-catenin pathways in microglia, inducing M2-like profile. This activation increases the invasion of GBM cells.	[67,68]

### 3.2. Non-Cellular Components

#### 3.2.1. Extracellular Matrix

The ECM comprises ~20% of the brain mass [75] and its function is not only to provide structural and biochemical support to surrounding cells, but to also play a key role in the regulation of several cellular processes, such as brain tissue homeostasis, viability, cell differentiation, migration, and invasion [75]. The brain’s ECM major components are hyaluronic acid (HA), tenascin-C (TNC), laminin, and collagen, among others, which have a key role in modulating invasiveness [76,77]. Infiltrative tumor cells are capable of remodeling and degrading ECM by MMPs released into the extracellular space [78]. MMP-2 and MMP-9 are the most highly expressed MMPs in GBM tissue, and they have been linked to a poor patient prognosis [79]. GBM cells are able to modify ECM components, promoting infiltration. Wiranowska et al. demonstrated in human U373 and mouse G26 GBM cell lines that the activation of CD44 by its ligand HA stimulates the synthesis and secretion of HA [80]. Additionally, Annabi et al. reported that CD44/HA interaction promotes glioma cell infiltration into healthy brain tissue by the upregulation of MT1-MMP expression [81]. Osteopontin (OPN), another component of the ECM, has been identified as a CD44 ligand, so its expression has been proposed as a possible modulator of GBM aggressiveness [82]. CD44/OPN have a perivascular expression pattern and their interaction induces the activation of the γ-secretase-regulated intracellular domain of CD44, which promotes migration, invasion, and stem cell-like phenotypes of GBM cells via CBP (CREB binding protein)/p300-dependent enhancement of HIF (hypoxia-inducible factor)-2α activity [82]. Furthermore, the co-expression of CD44 and OPN has been identified in GBM perivascular tissues and has been related to the stem-like phenotype [82].

Another ECM component is TNC glycoprotein, which has been implicated in embryogenesis, wound healing, and tumor progression [83]. It has been reported that elevated levels of TNC are correlated with GBM patients’ poor prognosis [84]. GBM cells produce and release TNC into the extracellular space, thereby changing the composition of the surrounding tissue, which facilitates the cellular invasion process [85]. Hirata et al. demonstrated that TNC silencing does not affect the proliferation of GBM cells, but impairs the in vitro cell migration in a two-dimensional substrate and decreases the cellular infiltration towards the brain parenchyma in a mouse xenograft model [86]. On the other hand, it has been reported that the downregulation of TNC in the TME causes less tumor cell infiltration but increases proliferation and the growth of the tumor mass. These results suggest that TNC could act as an important modulator of the so-called “go or grow” phenomenon [87,88]. This hypothesis proposes that adherent cells reversibly switch between migratory and proliferative phenotypes, where cells in the migratory state are more mobile than those in the proliferative state, because they are not using energy for proliferation [88]. Another study showed that MMP-12 is upregulated in GBM cells when they are exposed to a three-dimensional matrix enriched in TNC, and its silencing decreases in vitro cell invasion through MMP-12 downregulation [89]. Additionally, TNC could stimulate GSCs invasiveness by ADAM9 metalloproteinase expression and activity, via the c-Jun NH2-terminal kinase pathway [90]. The ADAM9 relevance in the invasive phenotype of GSCs was evaluated in histological samples of GBM patients and orthotopic xenograft models, reporting an elevated co-expression of ADAM9 and TNC in the invasive front of the tumor tissue [90]. In a differential microarray expression analysis, in which healthy tissue was compared with GBM tumor tissue, Ljubimova et al. identified two genes that were constitutively expressed in GBM, EGFR and laminin-α4 [91]. The α4 chain of laminin is constituted by laminin-9, laminin-8, or laminin-14 [92]. They found that during the GBM tumor progression, a switch of laminin-9 to laminin-8 occurs in the α4 chain of the blood vessel basement membranes [93]. In fact, laminin-8 silencing impairs in vitro cell invasion by almost 50% [93]. Furthermore, the study of 37 glial primary tumors, including 23 GBMs, demonstrated that the overexpression of laminin-8 was strongly associated with reduced recurrence time after surgery and poor survival of GBM patients, suggesting a role of laminin-8 in GBM spreading and infiltration [94]. Another research group analyzed 57 GBM biopsies reporting high expression of laminin-2 and -5 in infiltrative areas, compared with the tumor core, suggesting that laminin is involved in GBM invasiveness [95]. Similarly like laminin, brevican (BCAN) is a proteoglycan exclusive to the central nervous system and its expression is increased in GBM compared to healthy brain tissue, and has been associated with aggressiveness at the late stage of glioma progression [96]. Mice intrathecal inoculation with human GSCs primary cultures demonstrated that BCAN was expressed mainly in GSC niches, promoting tumor cells’ infiltration [97]. Moreover, Nakada et al. demonstrated that ADAMTS-5, a member of the ADAMTS family, is overexpressed in GBM cells and degrades BCAN, promoting cell invasion [98] (Figure 2).

Fibronectin (FN) is a glycoprotein present in ECM, which has been found highly expressed in human GBN samples, mainly in diffuse tumor tissue areas [99]. Serres et al. showed that FN silencing results in alterations of GBM ECM composition, enhancing the persistent directional migration of single cells, but dysregulating the collective cell adhesion and motility of spheroids through a laminin-rich matrix [100]. Additionally, FN impact in the adherence of GSCs and increased MMPs activity has been reported. FN promotes cell adhesion and differentiation of GSCs by activation of the adhesion kinase/paxillin/Akt signaling pathway [101]. Furthermore, FN can activate the axis Stat3-ODZ1-RhoA/ROCK, promoting invasion of GBM cells [102] (Figure 2).

Despite the low collagen expression in brain tissue, in GBM its levels are elevated, playing an important role in cell motility and invasion [103]. However, it is not just the composition of the ECM that impacts GBM invasiveness: the architecture of ECM is also important in these processes; indeed, invasive GBM tissue presents a disorganized collagen structure compared with focal GBM growth [104,105]. In addition, collagen is involved in stemness maintenance; specifically, type 1 collagen has been reported to be enriched in the perivascular niche of CD133(+) GSCs, enhancing invasiveness through integrin and PI3K/Akt signaling pathways [106]. All together, these results suggest that ECM components in GBM have important and essential roles in the invasive phenotype initiation, progression, and maintenance of cells, and are suitable targets for novel therapies (Table 2).

#### 3.2.2. Hypoxia

Low oxygen (O_2_) pressure is a main GBM TME characteristic and regulates several cancer hallmarks, such as chemoresistance, angiogenesis, cell proliferation, and invasion [107]. It has been reported that brain O_2_ levels oscillate between 12.5 and 2.5% (brain physioxia), but in tumor tissue, O_2_ levels decrease from 2.4 to 0.1% (hypoxia) [108]. GBM hypoxic regions have been largely described and have been proposed as a tumor marker for patient poor prognosis [108]. The GBM hypoxic microenvironment is enhanced by excessive tumor cell proliferation and deficient blood vessel formation [109]. GBM cells have the ability to adapt and persist in the hypoxic niche due mainly to the expression of HIFs, a family of transcription factors induced under low O_2_ pressure conditions [110]. HIFs are heterodimers composed of one oxygen-sensitive HIF-α subunit (HIF-1α, HIF-2α or HIF-3α), which dimerizes with the constitutively expressed HIF-1β. Under physiological O_2_ pressure, the α subunit is hydroxylated by prolyl-hydroxylases in two proline residues, promoting the binding of von Hippel–Lindau tumor suppressor (pVHL) leading to its ubiquitination and degradation [111]. However, under low O_2_ pressure such as the hypoxic GBM microenvironment, hydroxylation is inhibited, thereby preventing degradation, and promoting the α subunit stabilization. Alpha subunits translocated to the nucleus and dimerized with the HIF-1β subunit are able to regulate gene expression involved in angiogenesis, metabolism adaptation, and invasiveness [112]. In GBM it has been reported that subunits HIF-1α and HIF-2α are considered the main regulators of cancer aggressiveness, including the invasive phenotype [113], and the shifting of cellular metabolism from oxidative phosphorylation to glycolysis [114]. Immunohistochemistry analysis of human and murine GBM samples revealed that HIF-1α was mainly detected in pseudopalisading cells around necrotic areas and the infiltration front [115]. Results obtained by Fujiwara et al. showed that HIF-1α silencing in four GBM cell lines resulted in the reduction of cell infiltration into healthy brain tissue in an organotypic model [116]. In addition, under hypoxic conditions, increased expression of MMP-2 and MMP-9 was observed, but HIF-1α silencing downregulated MMPs’ expression/secretion in GBM cell lines [116]. A GBM cell microrray showed that several invasion-related genes, such as *ADAM-5* and *MAP4K4,* were upregulated in GBM GL261 under hypoxic conditions, but this was impaired under HIF-1α silencing [117]. In GSCs, Li et al. reported that HIF-2α was overexpressed in GSCs-CD133(+), which was associated with the regulation of genes related to survival, stemness, and cell invasion [118]. Moreover, Johansson et al. showed that GSCs-CD44(+) can stabilize HIF-2α under hypoxic conditions in a CD44-dependent manner, upregulating HIF-2α target genes [119]. In human samples and cell lines, hypoxia induces the expression of the complex plasminogen system (PAI-1), plasminogen receptor (S100A10), and Upa receptor (uPAR) at the cell surface, enhancing ECM degradation [120]. Furthermore, GBM hypoxia induces an EMT-like process, turning static polarized cells into migratory and invasive cells [121]. These phenotypic changes are accompanied by E-cadherin downregulation and N-cadherin upregulation, mainly by transcription factors such as twist, snail, and ZEB [122]. These transcription factors have hypoxia response elements at their gene promoter regions, so their expression is upregulated under hypoxia by HIF stabilization [122]. GBM human samples showed that ZEB1 and Twist1 are preferentially localized at the invasive edge and pseudopalisades/necrotic-associated hypoxic areas [123,124]. In GBM cells (U87MG and SNB75), in vitro migration is mediated by ZEB1 under hypoxic conditions, but it is reversed under digoxin (HIF-1α inhibitor) treatment, suggesting that this process is modulated by HIF-1α through ZEB1 expression [125]. Twist is another transcription factor that induces GBM invasion under hypoxia; in fact, twist mRNA levels are increased in GBM tissue in comparison with those in healthy tissue or lower-grade glioma, and their expression is associated with poor prognosis [126]. Mikheeva et al. found that Twist1 significantly increased GBM cell line invasion in orthotopic xenotransplants and increased expression of genes in functional categories associated with adhesion, ECM proteins, cell migration and actin cytoskeleton organization, promoting changes towards a mesenchymal phenotype with a switch in the expression of E- to N-cadherin [127]. Despite Twist1 being regulated by HIFs, hypoxia-independent mechanisms are also described; for example, p75 neurotrophin receptor (p75NTR) downregulation enhances Twist1-dependent GBM EMT and invasiveness [128].

Moreover, recent findings suggest that GBM hypoxia regulates gene expression in an HIF-independent way. For example, FAT (FAT atypical cadherin) has been reported as having a novel regulatory effect on EMT/stemness markers, independent of HIF-1. The functional relevance of FAT was published by Srivastava and colleagues, where the knockdown of FAT1 reduced migration and invasion capacity of GBM cells under hypoxic conditions [129]. The decrease of oxygen-dependent ten eleven translocation (TET) activity affects the methylation of aggressiveness-related genes involved in angiogenesis, motility, and cell invasion [130]. Furthermore, it has been suggested that hypoxia induces epigenetic regulation of the transmembrane protein odd Oz (ODZ1, also known as TENM1), altering DNA methylation status and activating the ODZ1-mediated migration of GBM cells [131].

Metabolic reprogramming allows GBM invasive cells to generate the energy necessary for colonizing surrounding brain tissue and adapt to hypoxic microenvironments with unique nutrient and oxygen availability [132]. Enhanced glycolysis has dominated glioblastoma research with respect to tumor metabolism. Nevertheless, global profiling experiments have identified roles for lipid, amino acid, and nucleotide metabolism in tumor invasion [132]. In this way, it has been demonstrated that expression of the fatty acid uptake channel CD36 is upregulated in hypoxia in several types of cancer [133]. Clinical data show that CD36+ cells correlate with greater invasion and poor prognosis in several carcinomas, and CD36 inhibition also impairs metastasis in breast cancer-derived tumors and human melanoma [133]. In GBM, CD36 is expressed in the cancer stem cells, and is associated with a pro-invasion phenotype [134]. However, more studies are required to elucidate the mechanism by which this channel drives invasion. On the other hand, the amino acid aspartate has been shown to be a limiting metabolite for GBM cellular proliferation under hypoxia, which has critical implications as these tumors typically outgrow their blood (and therefore oxygen) supplies quickly [135]. Overall, a rigorous understanding of the metabolic features that define invasive GBM cells could provide novel therapeutic targets.

Various microenvironments in GBM—the hypoxic core, the perivascular niche, and the invasive tumor edge—promote GSCs’ heterogeneity and dynamic behavior [136]. These stem cells hold the ability to adapt to metabolic changes in several tumor microenvironments [136]. GSCs persisting in the perivascular microenvironment present a different metabolic profile than those residing in the tumor hypoxic microenvironment. GSCs exhibit a proneural phenotype in the perivascular niche while they show a more invasive mesenchymal phenotype in the hypoxic core [136,137]. While GBM cells mainly express the glycolytic enzyme pyruvate kinase M2 (PKM2), which promotes aerobic glycolysis for glucose metabolism, GSCs instead express both pyruvate kinases: PKM2 and PKM1 [137]. PKM1 induces mitochondrial metabolism, allowing GSCs to switch between aerobic glycolysis and oxidative phosphorylation. While GSCs in the perivascular microenvironment rely primarily on glucose metabolism, glutamine dependency has been observed in the hypoxic mesenchymal GSCs [136]. GSCs’ metabolic changes are a challenge when choosing therapeutic targets, and the target of multiple pathways simultaneously should be considered [132]. For example, radiation targets highly proliferative cells, sparing slow cycling cells [136]; therefore, a combination of metabolic inhibition and radiation could be promising [132]. Understanding the GSCs’ role in tumor metabolism is a crucial step in understanding the complexity that GBM represents.

In addition to the molecular mechanisms described by hypoxia promoting cell invasion (Table 3), There are other mechanisms by which low oxygenation levels are able to maintain the invasive capacity of GBM cells, like the release of molecules into the extracellular space, such as adenosine (Figure 2).

#### 3.2.3. Adenosine

Adenosine has been recognized as one of the molecules that increase in the TME of GBM [138]. Adenosine is a purine nucleoside involved in physiologic processes such as immune cell function suppression [139], neuromodulation [140], and having an anti-nociceptive effect [141]. Adenosine concentrations under physiological conditions are ~30–200 nM; however, in pathological conditions (e.g., inflammatory or GBM tumor tissue), its concentrations increase to ~1–10 µM [142]. Adenosine is produced from adenosine triphosphate (ATP) hydrolysis to adenosine diphosphate (ADP), which is dephosphorylated to adenosine monophosphate (AMP) by ectonucleoside triphosphate diphosphohydrolase (CD39) [143]. Tumoral AMP can be further dephosphorylated to adenosine by 5′-ectonucleotidase (CD73) [144] and/or phosphatase acid prostatic (PAP) [145]. Additionally, extracellular adenosine concentrations are regulated by equilibrate (ENTs) and concentrative (CNTs) nucleoside transporters [146], or enzymes metabolizing adenosine such as adenosine deaminase (ADA) and/or adenosine kinase (ADK) [146].

Adenosine regulates biological processes through four receptors, A_1_AR, A_2A_AR, A_2B_AR, and A_3_AR. A_1_AR and A_2A_AR are higher-adenosine-affinity receptors, while A_2B_AR shows a relatively lower adenosine affinity. This assumes that A_2B_AR has major physio-pathological significance, where there is a dramatic increase in extracellular adenosine concentration [147,148]. In the case of A_3_AR, its affinity for adenosine is like that of A_1_AR and A_2_AR [149]. It is expressed ubiquitously in many tissues and has a low level of expression in the brain [150]. Adenosine production and signaling have a significant role in the development, progression, and aggressiveness of several cancers, including GBM [151,152,153,154]. Yan et al. reported CD73-dependent adenosine production promotes GBM aggressiveness. In fact, the tumor size of CD73-KO mice was significantly smaller and less invasive compared with that of CD73-WT mice. Interestingly, CD73 expressed only in mice ECs was much more invasive and caused complete distortion of brain morphology. In addition, these authors suggest that the absence of CD73 and invasiveness downregulation is mediated by decreased MMP-2 activity [155]. Recently it has been reported that CD73 regulates GBM cell migration and invasion through adenosine pathways in vitro and in vivo, which is associated with MMPs and vimentin expression [156]. Similar to the previous data presented, another group reported the reduction of CD73 protein decreases expression of EMT-related genes and MMP-2 activity, with a significant suppression of GSCs’ invasion [157]. Adenosine/A_3_AR increase MMP-9 mRNA and protein levels through the activation of ERK 1/2, JNK, Akt, and AP-1, thereby increasing GBM cell invasion [158]. Despite A_1_AR being a high-affinity receptor, some report suggests a role in progression of GBM. Synowitz et al. showed in A_1_AR–deficient mice a more aggressive GBM growth, with strong accumulation of microglial cells around the tumor tissue. In the same study, they observed that activation of the A_1_AR decreased the GBM activity of MMP-2, suggesting that A_1_AR plays an anti-tumorigenic role [159]. Extracellular adenosine is able to activate A_2B_AR in GBM cells, and this signaling is involved in the γ-radiation-induced increased DNA damage response and the enhanced migration ability and actin remodeling of human glioblastoma cell line A172 [160]. Recently, a study demonstrated the importance of A_2B_AR receptor signaling in the migratory/invasive capacity of glioblastoma stem cells (GSCs) under hypoxic conditions. Analysis of data from TCGA shows a correlation between A_2B_AR expression and high-grade glioma tumors and hypoxic necrotic areas. These results show that A_2B_AR levels increase in GSCs under hypoxic conditions, which increases their migratory and invasive capacity. When A_2B_AR is blocked, the migratory and invasive capacity decreases. In a mouse xenotransplantation model, it was shown that MRS1754 treatment does not affect tumor volume but can decrease blood vessel formation and VEGF expression. [161]. GBM neovascularization is essential for invasiveness and is also promoted by adenosine signaling through A_3_AR regulating EC marker expression, such as that of CD34, CD144, and vWF, and new blood vessel formation in vivo [162]. In turn, GSCs persist in a hypoxic microenvironment, which promotes extracellular adenosine production by PAP activity [145]. Liu et al. reported that HIF-2α induced PAP expression, enhancing adenosine production, which activated A_2B_AR, and this way promoted some stemness characteristics of GSCs, such as self-renewal under chronic hypoxia [145]. In addition, A_3_AR blockade under hypoxia decreases EMT marker levels and GSCs’ invasiveness [163]. Recently, new approaches were proposed for GBM patients’ treatment, which consist in adenosine’s degradation using recombinant adenosine deaminase (ADA) as a potential GBM target [164]. In fact, adenosine depletion using ADA dramatically decrease GSCs’ migration and invasion under hypoxic conditions by altering HIF-2α stabilization, thereby downregulating MMPs, Twist1, ZEB1, and snail expression [164]. The correlation between adenosine receptor signaling and HIFs’ expression regulation under hypoxia has been investigated in GBM. The involvement of A_3_AR signaling, specifically though the MAP kinase, mediated accumulation of HIF-1α under hypoxia [165,166]. In this regard, modulated adenosine signaling may represent a novel approach of treatment for GBM patients (Table 4 and Figure 2).

#### 3.2.4. Senescence and Glioblastoma

Life expectancy has increased due to medical and scientific advances, but age-related diseases such as cardiovascular diseases, neurodegenerative diseases, and cancer are also increasing [167]. Cancer is now the leading or second cause of death in many countries and is predicted to double its incidence by 2070 [167]. Glioblastoma is more common in those aged 75 to 84 and is related to age risk factors such as neuroinflammation, accumulation of senescent cells, and weakened immune system function [168].

The transcription factor NF-kB exhibits a stronger association with aging through regulation of immune responses, cell senescence, apoptosis, metabolism, and chronic inflammation, which are associated with most aged-related diseases [169]. The activation of NF-kB maintains persistent neuroinflammation in the GBM microenvironment through the expression of proinflammatory cytokines such as IL-6, IL-1β, and TNF-α, as well as chemokines (CCL2, CCL5, CXCL1) [170]. The increase in CCL2 and TNF-α by NF-kB can also recruit macrophages and microglia into the GBM TME, promoting tumor growth in vivo [171].

On the other hand, persistently senescent cells can induce tumor growth and/or promote immune suppression, favoring tumor progression, and a chronic inflammatory microenvironment [172]. In glioblastoma, induction of senescence using temozolomide has been shown to downregulate DNA repair pathways, potentiating temozolomide-induced damage [173]. In contrast, ionizing radiation (IR), another gold standard GBM treatment, can induce senescence in non-neoplastic brain cells surrounding the tumor area, and consequently, stimulate tumor growth, recurrence, and aggressiveness [174]. Therefore, is still unclear how regulation of cancer development is controlled by its interactions with the adjacent senescent cells in the TME.

As previously described, GBM cells can remodel their own cytoskeleton and ECM, in order to favor tumor invasion [175,176]. With this in mind, we note age-related ECM alterations, such as weaker elastin fibers and stiffness of collagen fiber, facilitate this process [177]. Engler et al. evidence a regulatory role of ECM stiffness in stem cells’ lineage specification [178]. Tumor cells sense relative ECM stiffness, inducing EMT and increasing their migratory and invasive capabilities, traits of mesenchymal cells, favoring a more aggressive phenotype [179,180]. In conclusion, there are several aging-related mechanisms that can stimulate a carcinogenic response, but their effects on cancer promotion and/or prevention remain unclear.

## 4. Patient-Derived Glioblastoma Organoids for Modeling of Tumor Microenvironment and Invasiveness

To understand comprehensively the interaction of cellular heterogeneity and pathways involved in glioblastoma invasiveness requires the use of clinically relevant models that embrace the complexity and the biology of parent tumors. Patient-derived glioblastoma organoids (PD-GBOs) have emerged as a more accurate and potentially feasible field in the study of this biology and as a preclinical ex vivo model [181]. PD-GBO models, unlike traditional in vitro models such as monolayer tumor sphere cultures (neurospheres), are able to maintain intra-tumoral heterogeneity and expression of key parental tumor genes [182,183,184].

The 3D PD-GBOs are cancer stem cell-derived cultures grown within extracellular matrix that capture the cellular and microenvironmental heterogeneity found in the primary tumor and enable the simultaneous culture of functionally and phenotypically diverse stem and non-stem glioblastoma cell populations [185,186]. In these 3D cultures, fresh tumor tissues resected during surgery are or not subjected to enzymatic digestion and are cultured in a defined medium with extracellular matrix. The first report of PD-GBOS from GSCs was described in 2016 by Hubert et al., who modified the original procedure of Lancaster et al. [187,188] and grew organoids from primary cultures in a suspension of 80% Matrigel and 20% Neurobasal complete media (Neurobasal medium supplemented with EGF, bFGF, B27, glutamine, sodium pyruvate, and antibiotics). Under these conditions, the organoids are established in 2 weeks, reaching a size of approximately 3 to 4 mm after 2 months, and can be stable and viable after more than a year of continuous culture without passaging. In addition to presenting a similar spatial distribution to GSCs as seen in patient tumors, they are also able to replicate tumorigenic characteristics of GSCs, such as resistance to the radiotherapy 

Furthermore, when the PD-GBCs were orthotopically transplanted into NOD scid gamma (NSG) mice, the tumors displaying histological features, hypoxic gradient, and a single-cells infiltrative phenotype found in the primary tumor from which they were derived. Recently, the same research group showed that their PD-GBCs recapitulated aspects of biological drug resistance to current standard-of-care therapy (combination temozolomide and radiotherapy) in clinical practice [189]. Interestingly, using the same protocol to generate PD-GBCs, Chadwich et al. showed that PD-GBCs expressing a balance of nestin (progenitor marker) and/or differentiated neural markers (GFAPO, GALC and TUJ1) remain heterogeneous when cultured in 4D-printed insert arrays. Further, they were capable of mimicking brain cells, showing the multiple subtype phenotypes and transcriptional heterogeneity reflected in GBM. Subsequently, using another PD-GBCs approach, Jacob and colleagues, 2020 [184,185] reported a robust method to rapidly generate glioblastoma organoids in a defined culture medium (without exogenous EGF/bFGF or extracellular matrix) directly from fresh tumor specimens without cell dissociation. In this protocol, the tumoral pieces were growing in ultra-low attachment plates with medium containing 50% DMEM: F12 and 50% of Neurobasal medium supplemented with Glutamax, non-essential amino acid solution (MEM-NEAAs), N2, B27 without vitamin A, 2-mercaptoethanol, and insulin. Under these conditions, they were able to establish PD-GBOs from a wide range of glioblastomas within 1–2 weeks, with a 91.4% overall success rate, and were cultured for more than 48 weeks while maintaining their phenotype. These 70 bio-banked PD-GBO resources were characterized by the retention of phenotypic and molecular inter-and intra- tumoral heterogeneity, recapitulated hypoxia gradient, microvasculature, and immune cell populations. Orthotopic transplantation of intact GBOs with efficient engraftment resulted in 92% (48 out 52) displaying various degrees of infiltration into surrounding mouse brain tissue. Interestedly, the responses of the PD-GBOs to pathway-target drug treatments were consistent with the mutational status and maintained specific mutant antigens, such as EGFRvIII expression, demonstrating the usefulness of the platform for testing personalized therapy and optimize CAR-T therapy. Recently, using this same organoid protocol, Darrigues et al. [190] evaluated the invasion-inhibitive effects of three anti-invasive compounds (NF-κB, COX-2, and tubulin inhibitors) in PD-GBOs embedded in 33% Matrigel and using macros for FIJI/Image J to quantify invasion from outer margin organoids. However, although PD-GBOs are an invaluable tool that mimics the heterogeneous populations, these models have some limitations. The PD-GBOs’ generation can be of poor quality and unrepresentative samples can influence the success rate and spatial heterogeneity characteristic of the tumors. For example, it has been reported that the tissue hypoxic time before the start of the protocol, scarce tumoral cells tissues, and prevalence of necrotic cells in the fresh tissue can influence the results [184,185,186]. Furthermore, the external growth factor added in the medium can mean that the long time and/or multiple passages of PD-GBOs diverge from those of primary tumors from which they were derived. For this reason, it is highly recommended to maintain cryovials in a biobank (for less than 2 months, according to Jacob et al. (2020)) and carry out the characterization and experiments within a short time of culture to enable reproducible results. Finally, another limitation of this model is the difficulty of preserving the interaction between PD-GBOs and endothelial and immune cells over time. To address this limitation, PD-GBOs can be co-cultured [184] or bioprinted on a chip [191] with multiple cell types or grown as an air–liquid interface protocol [192] that is able to maintain the microenvironment for almost 60 days. All this background information shows that the PD-GBCs platform has potential in the study of microenvironment interactions, invasiveness for preclinical identification of new drugs, and immunotherapy [181].

Other examples of progress using organoid-based in vitro models for GBM invasion study have occurred within the past 4 years. Two studies using neoplastic cerebral organoids derived have demonstrated that introducing through CRISPR/Cas9 the activation of HRas^G12V^ and simultaneous disruption of TP53 and MYC amplification, or other oncogenic mutations in cerebral organoids, can initiates tumorigenesis [193,194], providing a model for the study of the biological and functional mechanism of GBM formation and progression. Moreover, the direct co-culture strategy has been reported in the study of GSCs’ invasion in organoids, where the patient-derived glioma stem cells can spontaneously fuse with iPSC-derived brain organoids to form hybrid organoids, so-called GLICOs (cerebral organoid gliomas) [195]. Using this model, the authors showed that GSCs were able to invade and proliferate within the healthy brain organoids and form interconnecting microtubes, recapitulating the in vivo behavior of GBM. Using a similar approach, the GLICOs showed increased expression of genes, such as PAX6, normally found in forebrain neural stem cells; the gap junction protein alpha 1 (GJA1) coding for connexin 43, glypican-3 (GPC3), COLL4A5, glutathione S-transferase P, DLK1, and LAMA1 required for GBM network formation and invasion [196]. Moreover, Goranci-Buzhala et al. [197] established and compared three different methods of 3D invasive assay in brain organoids (Assay 1: Co-culturing GSCs plus iPSCs; Assay 2: GSCs as dispersed cells in organoids; and Assay 3: fusion of GSCs as a compact sphere in organoids). The authors proposed that the model of GSCs as dispersed cells in organoids was the closest to having in vivo relevance, with extending invasive protrusions and microtube-like structures, and anticipated that these assays should serve as a starting point to develop advanced GSC invasion studies, including suitable tumor microenvironments such microglia, astrocytes, and the blood–brain barrier, and to identify anti-invasive compounds. However, these data suggest that GBM cells from different patients vary in their invasive capacity, so further studies will be necessary to confirm the differences and similarities in invasive capacities using different PD-GBCs under the different invasion models proposed. Figure 3 shows a summary of PDO-GBOs’ establishment, the main features and challenges of these models, and the proposed competitive strategies for study of in vitro and in vivo invasion. PD-GBOs have emerged as a feasible preclinical tumor model that, despite their limitations, address the possibility of different translational and clinical applications, such as the study of new targeted pathways involved in invasiveness and the prediction of treatment response and/or prognosis in glioblastoma patients.

## 5. Conclusions

GBM infiltration is not a process that depends exclusively on the neoplastic cells, but also on the internal TME and the tumor-associated surrounding tissue. Although this has been known for quite some time, in this review we have compiled the most relevant information about the cellular and non-cellular components that interact with GBM tumor cells, proposing potential therapeutic targets and poor prognosis biomarkers. The most relevant cellular components are non-tumor glial cells and ECs, which can interact directly with GBM cells and/or release soluble molecules, triggering signaling pathways that promote cell invasion. Furthermore, we highlight the role of hypoxia, adenosine, and HIFs’ positive regulation, promoting EMT and MMP activation. These processes lead to ECM remodeling, regulating the interaction with GBM cells, specifically by HA, TNC, laminin, and collagen. These macromolecules have a fundamental role in invasion-related cell signaling, as well as in the mechanical support that allows this process. Finally, it is important to emphasize the presence of GSCs, which have an exacerbated invasive capacity and are mainly responsible for tumor recurrence. Three-dimensional PD-GBOs mimics the genetical and functional tumoral heterogeneity features and offer a unique opportunity for analyzing GSC–host tissue interaction and invasiveness. In conclusion, the understanding of cellular and non-cellular components using PD-GBOs could serve as an ex vivo preclinical platform to find new therapeutic strategies to improve GBM treatments and patient management.

## Figures and Tables

**Figure 1 ijms-24-07047-f001:**
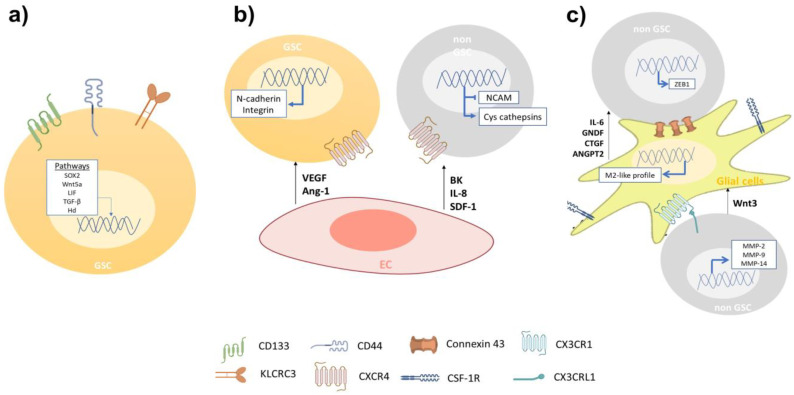
The migration and invasion of GBM cells is related to the inherent capacity and the cellular interactions that occur on the migratory front. (**a**) The invasiveness of GSCs is greater when it they are enriched in CD144, CD44, and *KLRC3* markers. Along with the above, the activation of the signaling pathways SOX2, Wnt5a, LIF, TGF-β, and Hh has been related to the invasive phenotype. (**b**) GBM cells infiltrate vascular areas and interact with ECs. These cells secrete factors such as VEGF, Ang-1, BK, IL-8, and SDF-1, which promote cell invasion by expressing adhesion proteins such as N-cadherin and integrin on GSCs and non-GSCs. (**c**) Glial cells secrete IL-6, GNDF, CTGF, and ANGPT2, which promote the invasion of GBM cells (not GSC). On the other hand, connexin 43, whose expression is elevated in astrocytes associated with gliomas, is an important gap junction protein that allows direct communication between astrocytes and GBM cells. Activation of CX3CL1/CX3CR1 in tumor-associated microglia/macrophages (TAMs) increases the adhesion/migration capacity of GBM cells through the expression of MMP-2, -9, and -14.Finally, the secretion of Wnt3 by non-GSCs towards microglial cells, induce the transition to profile like M2; this activation increases the invasion of GBM cells.

**Figure 2 ijms-24-07047-f002:**
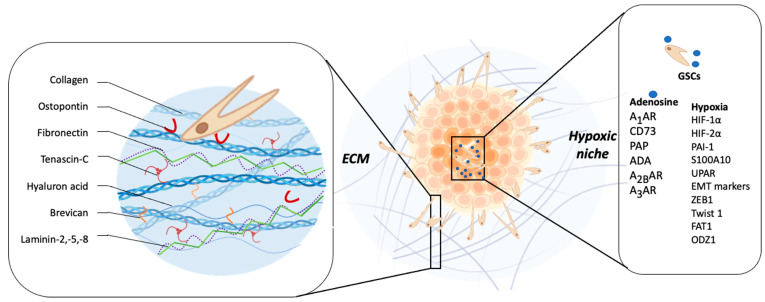
Relationship between tumor microenvironment components and GBM invasiveness. The main constituents of the brain ECM are collagen, osteopontin, fibronectin, tenascin-C, hyaluronic acid, brevican, laminin-2, -5, and -8, which have a role key in modulating invasiveness. The hypoxic microenvironment of GBM, characteristic of this tumor, promotes invasion mainly through the induction of one of the family of transcription factors induced under conditions of low O_2_ pressure (HIF-1 and 2), the increase in the expression of MMP-2 and MMP-9, plasminogen complex system (PAI-1), plasminogen receptor (S100A10), Upa receptor (uPAR), and markers of the epithelial–mesenchymal transition (EMT) process; expression and regulation of transcription factors such as ZEB1, Twist1, FAT1; as well as the induction of the epigenetic regulation of ODZ1. Adenosine has been recognized as one of the molecules that increases in the hypoxic niche of GBM. Proteins such as CD73, PAP, and ADA participate in its extracellular metabolism. This nucleoside regulates biological processes through the A_1_, A_2B_, and A_3_ adenosine receptors, ultimately promoting the invasive phenotype of GSCs and non-GSCs.

**Figure 3 ijms-24-07047-f003:**
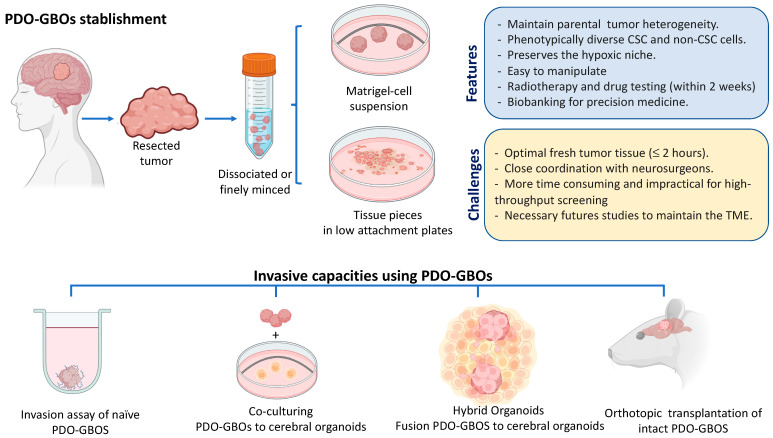
Patient-derived glioblastoma organoids (PD-GBOs) generation and models to study GBM invasiveness. The two mean models for establishment included resected fresh tumor samples, which are dissociated or finely minced and grown in Matrigel in Neurobasal complete medium (NBM) in the presence of exogenous growth factor (EGF/bFGF) [186], or finely minced (without-single cell dissociation) and grown on ultra-low attachment plates in a Neurobasal medium without exogenous growth factor alone to support the extracellular matrix from the tissue [184]. The main characteristics and challenges of these models are showed in the boxes at right. Below are shown the main models proposed for the study in vitro and in vivo of invasiveness and aggressive infiltration that have been used to study GSC invasion and that could be applied for the study of invasive ability of different PD-GBOs and screening of anti-invasive drugs. The figure was created with Biorender.

**Table 2 ijms-24-07047-t002:** Extracellular matrix components of GBM niches and their role in cell invasion.

Source	Molecule	Mechanism	Reference
**Extracellular matrix**	HA	Activates CD44, promoting U373 and G26 GBM infiltration.	[80,81]
	OPN	Activates CD44 by cleavage of its intracellular domain, promoting GBM migration and invasion.	[82]
	TNC	Downregulation in the TME causes less tumor cell infiltration but increases proliferation and tumor mass growth. It stimulates GBM cells, GSCs, and tumor front invasiveness by MMP12 and ADAM9 expression.	[84,85,86,87,88,89,90]
	Laminin-8	Its silencing impairs in vitro GBM cell invasion by almost 50%. Overexpression in tumors reduces recurrence time after surgery and poor survival of GBM patients.	[91,92]
	Laminin-2 and -5	High expression is associated with GBM-infiltrated areas.	[93,94,95]
	BCAN	Modulates ADAMTS-5 in GSCS, promoting ECM degradation and cell invasion.	[96,97]
	FN	Its silencing results in alterations of GBM ECM composition, dysregulating cell adhesion and motility. FN can interact with ODZ1, activating the RhoA/ROCK and JAK-STAT3 signaling pathway.	[98,99,100,101,102]
	Collagen	Is enriched in the perivascular niche of CD133+ GSCs, enhancing invasiveness through integrin and PI3K/Akt signaling pathways.	[103,104,105]

**Table 3 ijms-24-07047-t003:** Hypoxia and its mechanisms of action on GBM cell invasiveness.

Source	Molecule	Mechanism	Reference
**Hypoxia**	HIF-1α	Is detected in GBM infiltration front. Its silencing downregulates invasion-related MMPs and ADAM-5, MAP4K4 genes.	[115,116,117]
	HIF-2α	Is overexpressed in GSCs-CD133(+) and involved in gene regulation related to cell invasion.	[118,119]
	PAI-1, S100A10, uPAR	In human samples and cell lines, they are overexpressed at cell surface, enhancing ECM degradation.	[120]
	EMT markers	Are induced by HIFs and downregulate E-cadherin and upregulate N-cadherin, turning static polarized cells into migratory and invasive ones.	[121,122]
	ZEB1	Its expression is modulated by HIF1-α, is localized at the invasive edge in human tumor samples, and mediates migration in GBM cells.	[123,124,125]
	Twist1	Regulated by HIFs and p75NTR, promotes GBM invasiveness through ECM protein expression and cytoskeleton reorganization.	[126,127,128]
	FAT1	Promotes invasion of GBM cells through snail expression under hypoxia.	[129]
	ODZ1	Hypoxia induces its epigenetic regulation, altering DNA methylation status, activating GBM cell migration.	[131]

**Table 4 ijms-24-07047-t004:** Adenosine signaling pathway involved in GBM cells invasiveness.

Source	Protein	Mechanisms	Reference
**Adenosine**	CD73	Regulates GBM cell migration and invasion through adenosine pathways in vitro and in vivo, which is associated with MMP and vimentin expression regulation.	[155,156,157]
	A_3_AR	Increases MMP-9 mRNA and protein levels through activation of ERK 1/2, JNK, Akt, and AP-1, thereby increasing GBM cell invasion. A_3_AR blockage under hypoxia decreases EMT marker levels and GSCs’ invasiveness.	[158,162,163]
	A_1_AR	Decreases the GBM activity of MMP-2 and decreases microglia infiltration of GBM tissue.	[159]
	A_2B_AR	Signaling activation increases migratory ability through actin remodulation. Signaling activation induces MMP9 expression and activity in vitro, and increases VEGF expression in vivo.	[160,161]
	PAP	Activity is associated with stemness characteristics and infiltrative capacity of GSCs.	[145,163]
	ADA	Depletion using ADA dramatically decreases GSCs’ migration and invasion under hypoxic conditions by altering HIF-2α stabilization and downregulating MMPs, Twist1, ZEB1, and snail expression.	[164]

## Data Availability

Not applicable.

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
