# Peer review of "Glioblastoma Microenvironment and Invasiveness: New Insights and Therapeutic Targets"

_ijms, 2023, doi:10.3390/ijms24087047_

Round 1

Reviewer 1 Report

This is a very good and update review, with chapters of interest very well organized and written. Certainly it will be a reference for readers who need to find descriptions and discussion on the mechanisms involved in the glioblastoma microenvironment interaction in just one place.  However, pages 15 and 16 contain several typos and grammar errors and require English revision.

Minor:

Figure 1. Substitute Cx43 for Connexin 43 in the figure body to be in agreement with the respective legend (or substitute Connexin 43 for Cx43 in the legend).

p. 17 - line 580 - Please correct the figure number.

line 582 - "presence"

line 585 - Introduce the citation number.

line 606 - "serve"

References 61 and 120 - Journal name and year of publication: standardize the format.

Author Response

We sincerely appreciate the reviewer's valuable and constructive comments. We are committed to improving the quality of our work, and your observations have been of great help to us. Furthermore, we have made the changes indicated, so that you can make sure of these, we have highlighted them in red.

Rev 1

This is a very good and update review, with chapters of interest very well organized and written. Certainly, it will be a reference for readers who need to find descriptions and discussion on the mechanisms involved in the glioblastoma microenvironment interaction in just one place.  However, pages 15 and 16 contain several typos and grammar errors and require English revision.

Minor:

Figure 1. Substitute Cx43 for Connexin 43 in the figure body to be in agreement with the respective legend (or substitute Connexin 43 for Cx43 in the legend).

A: We change Cx43 for Connexin 43 in figure body

  1. 17 - line 580 - Please correct the figure number.

A: We designate the number 3 to the indicated figure. Thank you very much for pointing out the error.

line 582 - "presence"

A: We correct the indicated word. We leave it highlighted in red.

line 585 - Introduce the citation number.

R:  Thank you very much for pointing out the citation formatting error, we have changed the citation to the corresponding numbering.

line 606 - "serve"

A: We correct the indicated word. We leave it highlighted in red.

References 61 and 120 - Journal name and year of publication: standardize the format.

A: The references indicated were corrected according to the journal's standard format. Due to the addition of content suggested by reviewer 2, these citations changed their number, but they are marked in red so that you can check the correction we made.

Reviewer 2 Report

The manuscript entitled “Glioblastoma microenvironment and invasiveness: new insights and therapeutic targets” by Erices J and colleagues provided an integrative aspect of cellular interactions between TME including cellular and non-cellular compartments and the aggressiveness of GBM.   The authors have compiled an extensive body of literature and summarized it well in the form of tables and figures.  They also emphasized the use of glioblastoma organoids as a tool to study TMEs and the invasiveness of GBM. However, some important issues still need to be addressed before publication.  Some of the suggestions are listed below:

1. The section on tumor infiltrating immune cells and microglia should be described separately. The authors only mentioned the recruitment of peripheral macrophages by GBM cells. However, there are several cell types such as T cells and NK cells that also play a critical role in modulating the GBM microenvironment. This information needs to be included.

2. In addition to canonical microglial activation (M1 and M2 phenotypes), alternative microglial phenotypes have been discovered using RNA-seq analysis. There are several transcriptomic studies that provide crucial information about the response of TAMs to GBM, which is very important in this manuscript. It is recommended that the authors compile additional studies and discuss this information.

3. Although the authors have well described the effects of the hypoxic tumor environment on the alterations in gene expressions, metabolic reprogramming, termed the Warburg effect, also strongly influenced GBM invasion by generating the energy necessary for the process. The shift in metabolic pathways associated with GBM invasion should be discussed.

The authors have compiled the protocol for the generation of GBM organoids as a tool to study GBM holistically. It would be better if the authors could also state the limitations and the idea to improve the model as well as the translational application for use in clinical settings.

4. The incidence of GBM is rapidly increasing in the population over 65 years of age. How about adding the mechanisms of senescence and aging related to GBM in [3.2. Non-cellular components]?

Minor comments:

The reading of the entire manuscript by native English or an editing system is required.

1.     There are several punctuation mistakes, please correct.

2.     Typo: line 190, Hd -> Hh

3.     Line 192. IL-9 -> IL-8

4.     Typo: Figure 2 and line 340, lamin -> laminin

5.     Typo: line 443, mayor -> major

6.     Line 442 wrote that “The A1AR and A2AAR are higher adenosine affinity receptors”. However, Line “Despite, A1AR is low affinity receptor”. This might confuse readers. Please correct the sentence or consider the rewrite them.

7.     Typo: line 466, his -> this

8.     Typo: line 545, thar -> that

9.     Typo: line 553, ad -> and

10.  Line 232. Indentation is needed.

11.  In figure 2. Lamin-2, -5, -8 -> Laminin-2, -5, -8

12.  Line 340. Lamin-2, -5, -8 -> Laminin-2, -5, -8

13.  Line 489. Numbering is missing.

14.  Line 546. invasiveness  and -> invasiveness

Author Response

We thank the reviewer for taking the time to read our article and provide valuable feedback. Your comments and suggestions will be of great help to us in improving our work.

Revisor 2

The manuscript entitled “Glioblastoma microenvironment and invasiveness: new insights and therapeutic targets” by Erices J and colleagues provided an integrative aspect of cellular interactions between TME including cellular and non-cellular compartments and the aggressiveness of GBM.   The authors have compiled an extensive body of literature and summarized it well in the form of tables and figures.  They also emphasized the use of glioblastoma organoids as a tool to study TMEs and the invasiveness of GBM. However, some important issues still need to be addressed before publication.  Some of the suggestions are listed below:

The section on tumor infiltrating immune cells and microglia should be described separately. The authors only mentioned the recruitment of peripheral macrophages by GBM cells. However, there are several cell types such as T cells and NK cells that also play a critical role in modulating the GBM microenvironment. This information needs to be included.

A: Thank you very much for your comment, based on your indication an additional paragraph has been added describing some work on the role of NK cells and TILs in the GBM microenvironment. You can find this new information on page 7, lines 233 to 263 (Marked in yellow)

In addition to canonical microglial activation (M1 and M2 phenotypes), alternative microglial phenotypes have been discovered using RNA-seq analysis. There are several transcriptomic studies that provide crucial information about the response of TAMs to GBM, which is very important in this manuscript. It is recommended that the authors compile additional studies and discuss this information.

A: Thank you very much for this point, thanks to your comment, we also incorporated some work in our write-up on describing new activation "states" described through scRNA-seq based data analysis. This new text is on page 8, lines 284 -307. (Marked in yellow)

Although the authors have well described the effects of the hypoxic tumor environment on the alterations in gene expressions, metabolic reprogramming, termed the Warburg effect, also strongly influenced GBM invasion by generating the energy necessary for the process. The shift in metabolic pathways associated with GBM invasion should be discussed.

A: Based on your suggestion, we incorporated in the section describing the role of hypoxia on the invasive capacity of the WBC, a paragraph on the metabolic reprogramming induced by low levels of oxygenation of the WBC microenvironment. This is found on page 14, between lines 520-554 (Marked in yellow)

The authors have compiled the protocol for the generation of GBM organoids as a tool to study GBM holistically. It would be better if the authors could also state the limitations and the idea to improve the model as well as the translational application for use in clinical settings.

A: Thank you for the valuable suggestion to improve ours review. The main limitation of the organoid’s establishment has been included in page 18, lines 680-695. Other paragraph with the potential translational and clinical of this PD-GBOs implementation in page 19, lines 726-730

 The incidence of GBM is rapidly increasing in the population over 65 years of age. How about adding the mechanisms of senescence and aging related to GBM in [3.2. Non-cellular components]?

A: We incorporate a new section on the impact, cellular mechanisms associated with senescence and age. Focused mainly on neuroinflammation and changes in the characteristics of the ECM. This new section can be found on page 18, between lines 633 663 (marked in yellow)

Minor comments:

The reading of the entire manuscript by native English or an editing system is required.

  1. There are several punctuation mistakes, please correct.
  2. Typo: line 190, Hd -> Hh:
  3. Line 192. IL-9 -> IL-8:
  4. Typo: Figure 2 and line 340, lamin -> laminin:
  5. Typo: line 443, mayor -> major

A: We corrected the English errors; these are marked in yellow

  1. Line 442 wrote that “The A1AR and A2AAR are higher adenosine affinity receptors”. However, Line “Despite, A1AR is low affinity receptor”. This might confuse readers. Please correct the sentence or consider the rewrite them.:

A:Thanks for the comment, we changed the word low to high, because as indicated at the beginning of the section A1AR is a high affinity receptor.

  1. Typo: line 466, his -> this :
  2. Typo: line 545, thar -> that:
  3. Typo: line 553, ad -> and:

  1. Line 232. Indentation is needed.:

  1. In figure 2. Lamin-2, -5, -8 -> Laminin-2, -5, -8:
  2. Line 340. Lamin-2, -5, -8 -> Laminin-2, -5, -8:

  1. Line 489. Numbering is missing. : corregido

  1. Line 546. invasiveness  and -> invasiveness:

A: We corrected the English errors; these are marked in yellow.
